# Pluronic^®^ F127 Thermoresponsive *Viscum album* Hydrogel: Physicochemical Features and Cellular In Vitro Evaluation

**DOI:** 10.3390/pharmaceutics14122775

**Published:** 2022-12-12

**Authors:** Mariana S. Rocha, João V. C. Batista, Michelle N. O. Melo, Vania E. B. de Campos, Anna Lecticia M. M. Toledo, Adriana P. Oliveira, Paulo H. S. Picciani, Stephan Baumgartner, Carla Holandino

**Affiliations:** 1Multidisciplinary Laboratory of Pharmaceutical Sciences, Faculty of Pharmacy, Universidade Federal do Rio de Janeiro, Rio de Janeiro 21941-902, Brazil; 2Society for Cancer Research, Hiscia Institute, 4144 Arlesheim, Switzerland; 3Department of Pharmaceutical Sciences, Division of Pharmaceutical Technology, University of Basel, 4056 Basel, Switzerland; 4Department of Pharmacy, Universidade do Estado do Rio de Janeiro, Rio de Janeiro 23070-200, Brazil; 5Institute of Macromolecules Professora Eloisa Mano, Universidade Federal do Rio de Janeiro, Rio de Janeiro 21941-598, Brazil; 6Institute of Integrative Medicine, University of Witten/Herdecke, 58455 Witten, Germany; 7Institute of Complementary and Integrative Medicine, University of Bern, 3012 Bern, Switzerland

**Keywords:** mistletoe, thermal analyses, cytotoxic assays

## Abstract

*Viscum album* L., popularly known as mistletoe, is well known for its anti-cancer properties, and the pharmaceutical application of hydroalcoholic dry extracts is still limited due to its low solubility in aqueous media, and physicochemical instability. The Pluronic^®^ F127 is an amphiphilic polymer, which permits the solubilization of lipophilic and hydrophilic compounds. In this investigation, physicochemical features of hydrogel containing *V. album* dry extract (VADE-loaded-hydrogel) were performed by: dynamic light scattering (DLS), thermogravimetric analysis (TGA), differential scanning calorimetry (DSC), and transmission electron microscopy (TEM). VADE-loaded-hydrogel presented nanometer-size micelles with volume distribution ranging from 10.58 nm to 246.7 nm, and a polydispersity index of 0.441. The sample thermal analyses (TG and DSC) showed similar decomposition curves; however, the thermal events indicated an increase in thermal stability in relation to the presence of the extract. In addition to these interesting pharmaceutical features, IC_50_ values of 333.40 µg/mL and >1000 µg/mL were obtained when tumor (SCC-25) and non-tumor (L929) cells were incubated with VADE-loaded-hydrogel, respectively. The optical and ultrastructural cellular analysis confirmed the tumor selectivity since the following alterations were detected only in SCC-25 cells: disorganization of plasmatic membrane; an increase of cytoplasmatic vacuole size; alteration in the cristae mitochondrial shape; and generation of amorphous cellular material. These results emphasize the promising antitumoral potential of VADE-loaded-hydrogel as an herbal drug delivery system via in vitro assays.

## 1. Introduction

*Viscum album* L. has robust literature on its usage as a complementary treatment for cancer [1,2,3,4,5]. It is indicated at all stages of treatment against various types of cancer [6]. The aqueous extract of *V. album* has a cytotoxic action on tumor cells, improves the immune system response, and has an anti-inflammatory effect during cancer therapy [7]. The complementary treatment with *V. album* was able to reduce adverse effects in patients who were undergoing conventional treatment against cancer [8]. This reduction in adverse effects can lead to greater tolerance to conventional treatments and improve the patient’s quality of life [7]. The biological activity of this aqueous extract has been associated with a complex of compounds, such as viscolectins, viscotoxis, flavonoids, lignans, phenolic acids, and terpenes [9,10]. 

The non-aqueous extracts of *V. album* also have antitumor potential, which has been described in the literature of both in vitro [11,12,13,14,15,16,17] and in vivo experiments [18,19]. In previous studies carried out by our research group, the cytotoxic activity of hydroalcoholic extracts of *V. album* via 3-(4,5-dimethylthiazol-2-yl)-2,5-diphenyltetrazolium bromide (MTT) against tumor cells was reported [20]. This activity has been attributed to the presence of phenolic acids and flavonoids in significant amounts in such extracts [20,21,22]. Holandino et al. [22] quantified the total amount of viscotoxins (VT) and their isoforms A3 and B in the hydroalcoholic extract of *V. album* subsp. *abietis* from *Abies alba*. It is already known that the cytotoxic effect attributed to this polypeptide (isoform A3) is associated with the electrostatic interaction between this VT and cell membranes [23].

Furthermore, the metabolomic profile of different *V*. *album* ethanolic extracts was previously annotated by liquid chromatography, coupled to high-resolution mass spectrometry showing the presence of phenolic acids, organic acids, phenylpropanoids, flavonoids, terpenes, and sugars [24]. The multivariate analysis showed that the main compounds responsible for the *V*. *album* subsp. *abietis* differentiation were 6′-O-β-D-Apiofuranosylsweroside, Naringenin-pentose-hexose, 2-Hydroxy-4-(6-hydroxy-5,7-dimethoxy-4-oxo-4H-chromen-3-yl) phenyl 6-O-hexopyranosylhexopyranoside. Additionally, this hydroalcoholic extract of *V. album* subsp. *abietis* was able to reduce the glucose uptake and extracellular lactate in the human breast cancer cell line (MDA-MB-231). Therefore, the inhibition of important glycolytic enzymes, such as hexokinase, phosphofructokinase, and pyruvate kinase is an important antitumor mechanism triggered by *V. album* hydroalcoholic extracts since it is associated with the interruption of the Warburg effect [24]. 

Additionally, Batista et al. [25] reported 18.88 mg of chlorogenic acid per gram of VADE. According to the French Pharmacopoeia, chlorogenic acid is a chemical marker for quality control of *V*. *album* L. harvested from apple trees (*Malus domestica* Borkh). It is known that this phenolic compound from *V. album* ethanolic extract promoted cell death by apoptosis and cell cycle arrest in the C6 glioma cell line triggered by the reactive oxygen species [26]. 

Considering the promising antitumor activity of these secondary metabolites present in the alcoholic herbal extracts, and the challenges involved with the poor water solubility of their major constituents, a thermosensitive hydrogel containing VADE has already been developed [25].

The development of herbal medicinal products presents challenges due to the complex extract composition and poor water solubility of its major constituents. Poloxamer hydrogels show interesting thermosensitive properties with hydrophobic domains that help to retain poorly water-soluble compounds in the matrix. Poloxamer 407 is the commercial name for a class of non-ionic triblock copolymers composed of a central hydrophobic poly(propylene oxide) block with two hydrophilic poly(ethylene oxide) blocks on the sides (PEO-PPO-PEO) [27]. Poloxamer 407, trade name Pluronic^®^ F127, has a thermoreversible gelling behavior that occurs as a function of temperature in aqueous solutions with concentrations greater than 15% (*w/w*) [28]. It is widely used as an emulsifier, dispersing agent, solubilizer, stabilizer for oral and topical suspensions, wetting agent, and gelling agent in topical formulations [27]. 

Several scientific publications describe different applications related to the use of these copolymer blocks, such as: coating iron oxide nanoparticles [29]; thermal gelling agents in formulations containing clotrimazole [30]; and polymeric micelles loaded with oleanolic acid [31]. The intratumoral administration (in situ) of Pluronic^®^ F127 is an interesting pharmaceutical technology currently under evaluation for cancer treatment, which improves pharmacokinetic and pharmacodynamic properties, minimizes adverse effects, and enhances treatment efficacy [32]. According to Lin et al. [33], the in situ application of Pluronic^®^ F127 gel formulation provided high concentrations of the antineoplastic agent, both in 4T1 and MCF-7 tumor-bearing mice, in addition to prolonged retention of the drug inside the tumor [32].

It is well understood that an increase in temperature causes a decrease in the number of hydrogen bonds between the water and the hydrophobic segment (PPO) of the copolymer, leading to micelles formation [34]. Micellization is an endothermic process that occurs at temperatures of approximately 12 °C for a solution containing 20% (*w*/*w*) of Pluronic^®^ F127 [28,35,36,37,38], but this temperature may vary according to the copolymer concentration of the solution [28]. The micelles may also organize themselves into crystalline structures at high temperatures, forming a gel structure [35]. Gelation is also an endothermic process, but it needs a much lower amount of heat to occur when compared to micellization [35,37,38]. In this gel transition, the solution viscosity undergoes drastic molecular changes, which occurs rapidly at body temperature [33]. At higher temperatures, above the gel’s melting point, the gel cloud point occurs. This process involves decreasing the solubility of the hydrophilic chain (PEO) of the copolymer in water [39], leading to a gel phase separation [40]. 

The transition temperatures of the system can be strongly influenced by both the addition of excipients and drugs. Several studies have analyzed the influence of salts, organic solvents, and drugs on the micellization process, gelation, and in the cloud point of the Pluronic^®^ F127 solutions [36,40,41,42,43], using similar experimental tools also applied in the present study, such as: Dynamic Light Scattering (DLS), Transmission Electron Microscopy (TEM), and thermal analysis (TGA and DSC). According to Thapa et al. [36], the ability of Pluronic^®^ F127 to solubilize drugs will depend on its organization in solution—whether it is distributed as unimers (dispersed in the solvent individually) or as micelles. The presence of micelles can modify the pharmacological activity and stability of the drug. Therefore, in order to achieve an efficient controlled delivery drug system and good physical stability of the formulation, it is important to know its characteristics and the influence of its components in the system’s molecular organization [36].

The purpose of this study was to evaluate the in vitro physicochemical and antitumor effect of VADE in a polymeric delivery system of Pluronic^®^ F127 (VADE-loaded-hydrogel). The shape and size of micelles were analyzed in the presence and absence of VADE and excipients by DLS and TEM. The thermal decomposition of VADE-loaded-hydrogel micelles was evaluated by TGA, then DSC signals were recorded during the heating/cooling process to obtain a detailed behavior of hydrogels. The cytotoxic effects and the specificity of VADE-loaded-hydrogel to cancer cells were evaluated by in vitro assays on the tumor (oral tongue human squamous carcinoma—SCC-25 cells) and normal mouse adipose fibroblast (L929 cells). The morphological cellular aspects triggered by VADE-loaded-hydrogel were assessed by optical and electron microscopy. 

## 2. Materials and Methods

### 2.1. Plant Material

The botanical material consisted of berries, leaves and stems of *V. album* subsp. *ab-ietis* from the host tree *Abies alba* (A) and was harvested in February 2020, in St. Pantaleon, Switzerland. The botanical identification was carried out by Dr. Marcelo Guerra Santes and the voucher (CH Quaresma 18,328) was deposited in the Herbarium of the Faculty of Teacher Training of the Universidade Estadual do Rio de Janeiro, Brazil. The harvest pattern was conducted according to the previous methodology described by Holandino et al. [22] and Batista et al. [25]. 

### 2.2. Preparation the of Dry Hydroalcoholic Extract of V. album L.

*V. album* hydroalcoholic extract was obtained by maceration at room temperature followed by a daily agitation for 21 days, according to the methodology used by Holandino et al. [22] and described in the French and Brazilian Homeopathic Pharmacopoeias [44,45]. The plant material was fragmented and submitted to the total volume of the solvent in an environment protected from direct action of light and heat. After the maceration period, the extract was filtered. At last, the solvent was concentrated in a rotary evaporator (Büchi, Vacuum Pump V-700) under vacuum (Büchi 461 Water Bath) at 40 °C. The extract was frozen and subsequently dried by lyophilization (Christ Beta 2-8 LD) to obtain the *V. album* dry extract (VADE). 

### 2.3. Preparation of theThermosensitive Hydrogel

The hydrogel was prepared at a laminar flow and under sterile conditions according to the methodology adapted from Batista et al. [25]. Briefly, Pluronic^®^ F127 (20% *w*/*w*; Sigma-Aldrich, St Louis, MO, USA, CAS: 9003-11-6, Lot # BCBX9224) was dispersed until completely solubilized in sterile pure water for injections (Samtec Biotecnologia, Ribeirão Preto, SP, Brazil), and kept under refrigeration for 24 h. Subsequently, VADE (5% *w*/*w*), propylene glycol (5% *w*/*w*; Sigma-Aldrich, Rio de Janeiro, Brazil, Lot # DCBD0525V) and diethylene glycol monoethyl ether (5% *w*/*w*; Sigma-Aldrich, St Louis, MO, USA, CAS: 111-90-0, Lot # MKCD9083) were mixed in an ice bath (4 °C) to obtain the hydrogel (VADE-loaded-hydrogel). Aliquots of 1.5 mL were stored at room temperature in vials and were used in all experimental assays. As a control, a hydrogel without VADE (Vector Control) was prepared following the same methodology.

### 2.4. Size and Morphology of Hydrogel

Size distribution and polydispersity index (PdI) were determined by Dynamic Light Scattering (DLS) technique using Zetasizer Nano-S90^®^ (Malvern, UK). Hydrogels with and without VADE were diluted at 1:10 ratio in water for injections and analyzed at 25 °C. To assure a more reliable result, all the analyses were made in triplicate. The morphology of the formed micelles was assessed by Transmission Electronic Microscopy-TEM (FEI TECNAI SPIRIT BIO-TWIN). A drop of the sample was placed on a copper grid. Then, after 30 s, the solution excess was removed with a filter paper and a drop of 2.5% uranyl acetate solution was kept over the sample for 30 s. The samples were placed in a desiccator for 2 h before TEM analysis [46].

### 2.5. Thermogravimetric Analysis (TGA)

Thermogravimetric analysis was performed to evaluate the degradation characteristics and thermal stability of hydrogel without and with dry extract with a Thermogravimetric Analysis (TGA)—Model Q-500, TA instruments. Approximately 15 mg of each sample was measured at a heating rate of 10 °C/min from 30 °C to 700 °C under nitrogen atmosphere. Tonset and Tmax, as well as the residual material content, were determined. 

### 2.6. Differential Scanning Calorimetry (DSC)

Hydrogels’ thermograms were obtained by DSC Hitachi 7020 equipment. Approximately 10 mg of each sample was firstly stabilized at 2 °C for 5 min. After that, samples were heated to 60 °C at 10 °C/min, kept at 60 °C for 5 min, and then cooled from 60 °C to 2 °C (at 10 °C/min). This cycle was repeated one more time to obtain a detailed behavior of hydrogels within the range of temperatures evaluated. Simultaneously, a video record was done, and the most representative images obtained during the heating (from 2 to 60 °C) and cooling (from 60 at 2 °C) cycles were collected to evidence the pronounced visual macroscopic alterations. 

### 2.7. Flavonoid Content

The total flavonoids content in rutin equivalents was measured by ultraviolet spectroscopy (GENESYS 10S UV-Vis, Thermo Fisher Scientific Inc., Madison, WI, USA). The absorptions of rutin concentration series (5.0, 10.0, 15.0, 20.0, 25.0 and 30.0 μg/mL) were plotted to provide a linear calibration curve. The absorbance was determined at 360 nm and the measurements were performed in triplicate. Total flavonoids content of the extract was expressed as mg of rutin equivalents per gram of fresh material (mg/g FM) and mg of rutin equivalents per gram of dry extract (mg/g DE) [22,25,47].

### 2.8. Cell Lines and Culture Conditions

Mouse normal fibroblasts (L929) and oral tongue human squamous cell carcinoma (SCC-25) were obtained from the Rio de Janeiro Cell Bank (Duque de Caxias, Rio de Janeiro, Brazil).

L929 cells were cultured in Dulbecco’s Modified Eagle’s Medium (DMEM, Vitrocell Embryolife, Campinas, São Paulo, Brazil) and SCC-25 cells in a 1:1 mixture of Dulbecco’s Modified Eagle’s Medium and Ham’s F12 Medium (DMEM/F12). Both full mediums were supplemented with 10% fetal bovine serum (FBS, Vitrocell Embryolife, Campinas, São Paulo, Brazil) and penicillin (100 Ui/mL, Sigma-Aldrich, Merck KGaA, Darmstadt, German) and streptomycin (100 µg/mL, Sigma-Aldrich, Merck KGaA, Darmstadt, German). Cells were kept in a humidified atmosphere at 37 °C with 5% carbon dioxide (CO_2_).

### 2.9. Cell Viability Assay

Cell viability was evaluated by 3-(4,5-dimethylthiazol-2-yl)-2,5-diphenyltetrazolium bromide (MTT). Cells were seeded (1.125 × 10^4^ cells per well) in a 96-well plate and incubated for 24 h. Then, they were treated with VADE-loaded-hydrogel dissolved in full medium at the following concentrations of VADE: 100, 250, 500, 750 and 1000 µg/mL. As control groups, cells were incubated in culture medium (control cells) and with hydrogel vehicle (vector control, VC). Regarding to the vehicle control concentration, the cells were incubated with the highest concentration of hydrogel (20 mg/mL) previously evaluated in tumor and non-tumor cells [25]. After 24 h of treatment, the supernatant was discarded and 180 µL of each full medium and 20 µL of MTT solution (5 mg/mL) were added. Cells were incubated for 3 h at 37 °C, 5% CO_2_ in the dark. Lastly, they were centrifuged, and the formazan precipitate was dissolved in 200 µL of dimethylsulfoxide (DMSO) for reading in a microplate reader (Thermo Plate, TP-Reader) at 490 nm [48,49].

The half maximal inhibitory concentration (IC_50_) was calculated and expressed in μg/mL of the VADE. The results represent 3 independent experiments performed in quintuplicate [50].

### 2.10. Morphological Changes by May-Grunwald-Giemsa Staining

SCC-25 and L929 cells (4.5 × 10^5^ cells/well) were cultured in a 12-well plate, on a glass coverslip for 24 h. Subsequently, cells were incubated with a hydrogel solution (equivalent to 250 and 500 µg/mL and 500 and 1000 µg/mL) of VADE for SCC-25 and L929, respectively) in full medium for 24 h. The hydrogel without VADE was also eval-uated as a control using the highest concentration of hydrogel (20 mg/mL) previously evaluated [25]. Then, each well was washed twice with phosphate-buffered saline (PBS) and the cells were fixed with Bouin’s solution for 5 min. Afterwards, the cells were washed with 70% ethanol to remove all fixative solutions and with distilled water to remove the residual alcohol. Cells were dyed with Giemsa (Merck KGa, Darmstadt, Germany) and after 3 h the glass coverslips were immersed in a Petri dish containing 20 mL of distilled water acidified with 3 drops of glacial acetic acid (Merck KGa, Darmstadt, Germany). Further, the coverslips were placed on filter paper to remove the excess of water. The cells were submitted to the dehydration process, in which all of them were incubated for 10 s in different solutions of acetone (Reagen, Rio de Janeiro) and xylene (LabSynth, Diadema, São Paulo, Brazil). The cellular morphological features were examined under an optical microscope (Axioplan 2, Carl Zeiss, Göttingen, Germany) with a 20× objective. All images were obtained using a digital camera (Axi-ocam MRc) attached to the microscope [48,51].

### 2.11. Ultrastructure Alterations

The ultrastructure features were evaluated by transmission electron microscopy (TEM). SCC-25 and L929 cells (4.5 × 10^5^ cells/well) were cultured in a 12-well plate for 24 h. Subsequently, they were incubated with a hydrogel solution and with their re-spective controls in full medium (equivalent to 500 µg/mL of dry *V. album* for SCC-25 and L929 cell lines). After 24 h, treated and untreated cells were centrifuged, the supernatant discarded, and the cells were fixed for 1 h in glutaraldehyde 2.5% containing sodium cacodylate buffer 0.1 M (pH 7.2). Subsequently, the material was washed in the same buffer, post fixed in osmium tetroxide 1% for 1 h, and washed with 0.1 M cacodylate buffer. Finally, all cells were dehydrated in acetone series and the material was embedded in Polybed 812. Ultrathin sectioning was performed on an EM UC6 microtome (Leica Microsystems GmbH, Wetzlar, Germany). Samples were recovered on 300-mesh copper grids (Ted Pella, Inc.), stained with uranyl acetate and lead citrate, and observed on an FEI Morgagni transmission electron microscope (FEI Company, Hillsboro, OR, USA) operating at 80 kV.

### 2.12. Statistical Analysis

The statistical analyzes were performed using GraphPad Prism 7.0 software and differences were considered significant when *p* < 0.05.

## 3. Results and Discussion

### 3.1. Size and Morphological Characterization

Hydrogel formulations were successfully obtained by the “cold” method. Pluronic^®^ F127 in an aqueous solution at a critical micellar concentration (cmc) spontaneously forms spherical micelles [28,35]. The size distribution and morphology of the micelles formed were analyzed by DLS and TEM (Figure 1). 

DLS measurements of Pluronic^®^ F127 water solution have shown a micellar narrow-size distribution (Figure 1D). The hydrogel dispersed in water generated micelles with a size of 7.55 nm (100% of the micelles population) of hydrodynamic diameter (h.d) and a polydispersity index (PdI) of 0.223. The literature reports diameters of 10.6 nm [52] for Pluronic^®^ F127 water solutions, similar to the value found in our analysis. Additionally, hydrogel without extract (Figure 1E) showed similar micelles characteristics, in which 100% of the micelles population presented a size of 7.24 nm and PdI of 0.321 values. This result indicated that the hydrogel excipients (diethylene glycol and propylene glycol) did not significantly influence the micelles distribution parameters.

However, the addition of VADE to the hydrogel induced a different micelle volume distribution, which was clearly related to the high PdI value (0.411) and to the heterogeneous micelle volume distribution (Figure 1F). The percentage of volume distribution shows three different populations, in which 76.6% of Pluronic^®^ F127 micelles presented 10.58 nm (h.d), 22.6% presented 33.09 nm (h.d), and 1% presented a higher h.d value (246.7 nm). These results suggest the influence of excipients and the dry extract on the features of the micelles. 

The TEM morphological aspects of VADE-loaded-hydrogel confirmed the presence of larger micelles with greater negative contrast at the sphere periphery, probably due to the VADE distribution (Figure 1C). The Pluronic^®^ F127 water solution (Figure 1A) and hydrogel without VADE (Figure 1B) originated similar and uniform micelles structures. VADE-loaded-hydrogel presented a bimodal population (Figure 1C), which could be caused by aggregates, visible in Figure 1F. These aggregates should be further investigated. To better understand the size and morphological variations, TGA and DSC hydrogel thermal analyses were carried out.

### 3.2. Thermal Analysis of the Hydrogel

In the pharmaceutical area, thermal analysis techniques such as Thermogravimetric Analysis (TGA) and Differential Scanning Calorimetry (DSC) have been used for thermal characterization and determination of the stability of the pharmaceutical product [53,54]. Additionally, they have served as routine methods for tracking drug-pharmaceutical excipient interactions; they are very useful for polymer analysis [55]. The thermal behavior of hydrogel without VADE and the VADE-loaded-hydrogel was evaluated by the above-mentioned techniques. The thermal degradation profiles of the hydrogels without VADE (black curve) and VADE-loaded-hydrogel (green curve) showed small differences related to the presence of the extract, as seen in Figure 2.

A high percentage of samples’ mass loss occurred between 30–143 °C. The loss was approximately 70% and 75% of the weight, regarding the VADE-loaded-hydrogel and the one without VADE, respectively. This thermal degradation is mostly associated with the water loss of the polymeric hydrogel matrix. As reported in the literature, temperature ranges from 25 to 140 °C are associated with water loss in thermogravimetric analysis [55,56,57,58,59], and this large reduction in water-related mass is consistent since water is the most important ingredient of hydrogel formulations. However, at 128.6 °C, the behavior of the curves slightly changed, and, at around 140 °C, the hydrogel without VADE lost approximately 5% more of its mass, when compared to the VADE-loaded-hydrogel (Figure 2; circle 1). This result is compatible with the hydrogel composition since the VADE-loaded-hydrogel contains 5% *w*/*w* of VADE, which replaces 5% of water in the hydrogel without VADE. 

Additionally, the hydration of PEO and PPO blocks of Pluronic aqueous solutions can be strongly influenced by the presence of excipients and drugs [60]. The chemical composition of VADE, which includes phenolic compounds and flavonoids [20], may also be involved with PPO/PEO block bonds, promoting greater thermal stability to VADE-loaded-hydrogel. 

In the thermal curve of the VADE-loaded-hydrogel, a gradual loss of mass is observed in the temperature ranging from 143 to 350 °C. This mass loss might be associated with the degradation of VADE compounds. According to Souza et al. [61], in the range of 100.50 to 357.43 °C, organic extract compounds presented in a dry extract of *O. ficus-indica* suffer chemical degradation with a maximum loss at 357 °C, which is compatible with our results. On the other hand, at the same range of temperature, no mass loss was observed in the absence of extract (Figure 2, black curve). Therefore, we could infer that within this range of temperature, the slow and gradual weight loss was caused by VADE degradation.

The second major degradation event took place from 350 to 410 °C, and both curves are associated with the polymer thermal decomposition (Figure 2; black and green curves), corroborating the previous results described by Nguyen et al. [57]. The decomposition of samples began to occur at different temperatures: 346 °C in the absence of VADE, and 365 °C for the VADE-loaded-hydrogel (Figure 2; circle 2). It is possible to calculate the difference of temperature in this decomposition process, which was around 19 °C, which probably occurred due to a chemical interaction between PPO/PEO blocks and the remaining phytochemical substances of VADE, reinforcing the greater thermal stability of VADE-loaded-hydrogel. 

At the end of the thermal curve of the VADE-loaded-hydrogel (Figure 2; green curve), a minimum content of mass, of approximately 1.37% *w*/*w* (Figure 2; arrow), was detected in the temperature ranging from 410 to 500 °C. This residual content may be associated with the presence of inorganic materials, as previously detected by Souza et al. [61], which avoided the total thermal decomposition, even at the maximum temperature used (at 500 °C; Figure 2; green curve). On the contrary, the hydrogel without VADE (Figure 2; black curve) was almost entirely degraded, since a residue of 0.21% *w*/*w* (Figure 2; arrow) was detected by the end of the TGA.

The DSC analyses showed that both samples presented similar endothermic profiles, up until 15 °C (first heating cycle), then they started to gradually differentiate up to 40 °C, as observed in the thermogram (Figure 3). From this temperature, up to 55 °C, a greater difference in energy absorption was observed. Under our experimental conditions, from 57 °C to 58.2 °C, the curve of hydrogel without extract (Figure 3; black curve) presented an accentuated decrease because of its greater heat capacity (−22,500 mW; Figure 3) in comparison with VADE-loaded-hydrogel (−16,000 mW; Figure 3). According to Barba et al. [37], the main event that occurs in the heating curve of a Pluronic^®^ F127 solution (20% *w*/*w*) is the water evaporation process. It is an endothermic event that probably requires a greater amount of energy to break the water hydrogen bonds. These DSC results are consistent with the TGA analysis, since, as above mentioned, the VADE-loaded-hydrogel has less than 5% (*w*/*w*) of water in comparison to hydrogel without VADE. 

The simultaneous video recording showed important physical alterations of the samples during DSC experiments (Appendix A). It was possible to observe that during heating from 15–55 °C (Figure 4A; 15.0, 40.0, 50.0, and 55.0 °C) the samples did not present significant physical alterations. However, in the temperature range from 55 °C to 58.2 °C, drastic changes in the hydrogel characteristics were registered (Figure 4B): both hydrogels (with and without VADE) were subject to gradual phase transitions due to the increase in the temperature (at 55.0, 57.0, 58.0 and 58.2 °C; Figure 4). However, in the absence of VADE, the hydrogel suffered a disorderly deformation in which two phases were observed at the end of heating (Figure 4B; top). In contrast, the presence of VADE promoted a circular spatial organization of the sample, with a clear greenish center part surrounded by a dark-colored halo (Figure 4B; bottom). 

As described by Pandit and Kisaka [40], this physical alteration of thermoreversible gels is related to the gel cloud point (T_cp_), which promotes two generation phases: one of them with a high concentration of polymer (milky gel), and another phase with a low concentration of polymer (clear part) [40]. To understand what leads to a cloud point by increasing the temperature, the review from Almgren, Brown, and Hvidt [39] elucidated the behavior of the PEO homopolymer in an aqueous solution. Two theories were considered, one directly related to the breakdown of hydrogen bonds between water and polymer due to the increase in temperature, thus favoring the bonds between the polymers. The second one involves the conformation of the polymer chain based on possible rotations around the C-C bond of the -OCCO- structure, present in the PEO polymer. Nonpolar conformations become predominant in polymeric chains with increasing temperature, disfavoring polar bonds with water. These theories support the low solubility of the PEO polymer in aqueous solutions at high temperatures, leading to the separation of the gel into two phases (cloud point).

Furthermore, the literature reports that for water solutions of Pluronic^®^ F127, cloud point occurs at temperatures around 110 °C [43]. However, the transition temperatures can be reduced in the presence of other formulation components, such as salts [40] and organic solvents (glycerol). Alexandridis and Yang [62] showed a comprehensive picture of the influence of cosolvents in the self-assembly of polyether block copolymers (PEO-PPO-PEO), promoting micellization at lower temperatures. This behavior could be attributed to the competition between glycerol and the hydrophilic region of Pluronic^®^ F127 (PEO blocks), which could be reduced by the amount of water involved with PEO hydration [62]. In this study, the influence of propylene glycol (5% *w*/*w*) and diethylene glycol monoethyl ether (5% *w*/*w*) in the physical alterations of the hydrogel samples during DSC experiments could be confirmed, by the comparison of these results, with those obtained with a Pluronic^®^ F127 hydrogel at 20% (*w*/*w*) in water. Among other physical alterations, the most important one was the absence of cloud point when this polymer solution was evaluated (S2). 

In the present study, the hydrogel formulations have, in addition to Pluronic^®^ F127 two co-solvents, propylene glycol (5%) and diethylene glycol (5%). Considering that the cloud point might suffer the influence of co-solvents, a reduction of the hydrogel cloud point temperature might be expected as per the experimental result (Figure 4B). 

Figure 5 presents the heat flux during cooling as a function of temperature (10 °C/min). The cooling curves of both formulations showed exothermic peaks, and this thermic transition corresponds to the micelle melting process, which occurs with the reduction of the temperature of Pluronic^®^ F127 at 20% (*w*/*w*)[30,41].

The melting process can be analyzed by the onset temperature (T_onset_), the peak temperature (T_peak_), and the endset temperature (T_endset_). The hydrogel without VADE showed T_onset_ at approximately 20 °C, T_peak_ at 15 °C, and T_endset_ at approximately 10 °C (Figure 5, black curve). The effect of VADE presence on these transitions is evident in the exothermic peak shift to lower temperatures (Figure 5; green curve), and the following results were obtained: T_onset_ at approximately 12 °C, T_peak_ at 5 °C, and T_endset_ at approximately 2 °C. 

DSC studies performed with a solution of Pluronic^®^ F127 at 20% (*w*/*w*) found similar values for this exothermic peak. Shriky et al. [28] reported a temperature range from 18 °C to 11 °C for the exothermic peak during the cooling process. Another study using Micro-DSC found values very close to our present work, with Tonset = 20 °C, T_peak_ = 15 °C, and T_endset_ = 12.5 °C [35].

Additionally, the thermograms revealed an increase in energy enthalpy (∆H) associated with the presence of VADE: −5.42 mJ/mg (VADE-loaded-hydrogel) and −4.04 mJ/mg (hydrogel without VADE), confirming the greater thermal stability of this sample, since a significant amount of energy was required to promote the exothermic transition. 

Respective images of samples were recorded by a simultaneously recorded video (S1 and S3) during the observed exothermic event (Figure 5). The physical alterations were observed during cooling from 60 to 0 °C, and the images related to T_onset_, T_peak_, and T_endset_ were plotted for both formulations (Figure 6: Up-hydrogel without VADE; down-VADE-loaded-hydrogel). The samples comparison clearly shows the influence of temperature in the decrease of viscosity and also, as detected during DSC analysis (Figure 5), the presence of extract postponed the VADE-loaded-hydrogel micelle melting. The analysis of the specific temperature of each event showed different peaks with T_onset_ = 12 °C, T_peak_ = 5 °C and T_endset_ = 2 °C for VADE-hydrogel-loaded, and T_onset_ = 20 °C, T_peak_ = 15 °C and T_endset_ = 10 °C for hydrogel without VADE. This difference was accompanied by the physical alterations detected in these images (Figure 6 top and bottom). The analysis of the specific temperature of each event showed different viscosities related to each sample, VADE-hydrogel-loaded and hydrogel without VADE. Since Pluronic hydrogel is a non-Newtonian fluid, the decrease in temperature promoted a substantial decrease in gel viscosity, as previously described by our group through a rheological analysis of VADE-loaded-hydrogel [25]. These viscosity alterations were confirmed by video recordings (Appendix A). 

Thermal analysis showed important aspects of the formulation in the pharmaceutical field, such as the hydrogel degradation profile and the influence of excipients on thermal transition temperatures. The TGA results showed that the extract incorporated into the hydrogel starts its thermal degradation process at high temperatures (143 °C). Furthermore, the extract was also able to delay the thermal degradation of the polymer when compared to the vector control (hydrogel without VADE). The thermal characterization by DSC showed the influence of the excipients propylene glycol (5% *w*/*w*) and diethylene glycol monoethyl ether (5% *w*/*w*) in the cloud-point temperature reduction. This thermal event can influence the release of extract components since it alters the binding profile of the hydrophilic blocks of the polymer leading to phase separation of the gel. In addition, the DSC analysis also showed the absence of physical changes in the hydrogel at body temperature, in which the hydrogel would be exposed for possible applicability. 

### 3.3. Flavonoid Content in VADE-Loaded-Hydrogel

Total flavonoids content in VADE-loaded-hydrogel was estimated as mg/g of fresh material (FM) and as mg/g of dry extract (DE), by colorimetric assay, using rutin (RE) as standards (mg RE/g). The VADE-loaded-hydrogel presented flavonoid yields of 7.02 ± 0.84 mg/g FM and 44.97 ± 5.40 mg/g DE in rutin equivalents. The results of this study are in accordance with the range content of flavonoids described in the literature. Batista and collaborators [25] found a flavonoid concentration of 31.38 mg/g DE, as an equivalent of rutin when a dry extract from *V. album* subsp. *abietis* harvested in 2019 was analyzed. Stefanucci et al. [63] observed that the *V. album* L. from scots pine had a flavonoid content ranging 2.35–23.83 mg/g (rutin), depending on the plant part used (leaf, fruit, and seeds) and the extractive technique applied. These authors demonstrated that the leaves presented higher flavonoid content than the fruits and seeds. Another previous study conducted by our group registered the influence of the subspecies and the host tree in flavonoid amounts. It was observed that the *V. album* subsp. *album* from *Quercus* sp. harvested in the summer presented 9.67 mg/g FM of flavonoids, while the *V. album* subsp. *abietis* from *Abies alba* had 5.25 mg/g FM [22]. Therefore, the harvest season, subspecies, and host tree are important sources of information for the potential reproducibility of future studies.

### 3.4. Cytotoxic Activity of the Hydrogel Containing Dry Extract of Viscum Album

The literature reports the antitumor potential of non-aqueous extracts of *V. album* against tumor cells in both in vitro [11,12,13,14,15,16,17] and in vivo studies [18,19]. The cytotoxic activity of ethanolic extracts of *V. album* in tumor cell lines B16F10 (murine melanoma), K562 (human chronic myelogenic leukemia) [20], Yoshida (mouse sarcoma), MOLT-4 (human acute lymphoblastic leukemia) [22,25], and MDA-MB-231 (human breast cancer) [24] was previously reported by our group. The phenolic acids and flavonoids detected in these *V. album* extracts are involved with the cytotoxicity detected in these different tumor cell lines [20,22,24]. 

To investigate the cytotoxic activity of VADE-loaded-hydrogel, a colorimetric assay (MTT) was performed in a tumor cell line (oral tongue human squamous cell carcinoma– SCC-25) and non-tumor cell line (mouse fibroblast—L929). All statistical analyzes were performed in comparison to the control group (untreated cells). Figure 7 shows the cell viability of SCC-25 and L929 cells after 24 h of treatment with VADE-loaded-hydrogel dissolved in full medium, at different concentrations of VADE (100, 250, 500, 750, and 1000 µg/mL). The percentage of viable cells incubated with hydrogel without VADE (vector control—VC) shows the absence of toxicity in the normal cell line (L929) (*p* > 0.05; Figure 7B). Considering the cell viability for the non-tumor L929 cell line (Figure 7B), a significant increase in cell proliferation was observed with VADE-loaded-hydrogel from 100 µg/mL to 750 µg/mL (*p* < 0.05). However, at 1000 µg/mL, the cell viability was similar to the control. This interesting effect of VADE-loaded-hydrogel on fibroblast proliferation (Figure 7B) was also detected by Luu et al. [52] using Pluronic^®^ F127 in the nanoencapsulation of *Chromolaena odorata* extracts. The concentration-dependent cytotoxicity was based on Pluronic^®^ F127 micelles and, as detected by our results, presented no damage on human fibroblast cells.

The tumor cell line (SCC-25) presented a completely different profile, and all concentrations showed significant differences (*p* < 0.01) when compared to the control group (Figure 7A; *p* < 0.01). A dose-dependent response was detected, and the lowest concentration (100 µg/mL) decreased the cell viability by approximately 20%, followed by 250 µg/mL (40% of reduction). The higher concentrations, from 500 µg/mL to 1000 µg/mL, reduced the viability by more than 80%. The vector control showed low cytotoxicity and decreased the cellular viability by approximately 20% in comparison to the control group (*p* < 0.01). Previous studies carried out by our group evaluated the cytotoxicity of two hydrogels by WST-1 colorimetric assay [25]. Both formulations, one containing VADE and the other containing aqueous extract of *V. album* subsp. *abietis*, promoted cytotoxicity against tumoral cells (Yoshida and MOLT-4) in a dose-dependent manner [25].

The IC_50_ for SCC-25 and L929 cell lines were 333.40 and >1000 µg/mL, respectively, highlighting the higher sensitivity of SCC-25 cells when compared to L929 cells. Sárpataki and collaborators [64] also described a greater antitumoral effect in human cervical carcinoma (HeLa) cells using alcoholic *V. album* extract, and as detected in our results, minimal effects of the extract were observed in normal human fibroblast (Hfl). 

The subspecies, host trees, harvest seasons, and methodologies of extraction are factors that directly influence the chemical composition of the *V. album* extracts [10,20,21,22,25,65] and, therefore, its antitumor potential [10,20,21,22,25,65]. Melo et al. [24] showed that summer *V. album* mother tinctures were more cytotoxic than the winter preparations in breast cancer cell line (MDA-MB-231). Moreover, this study also demonstrated the interruption of the Warburg effect as a mechanism of action of these hydroalcoholic solutions [24].

These promising antitumoral effects of *V. album* ethanolic preparations motivated the use of the thermoresponsive Pluronic^®^ F127 hydrogel in which its attractive properties, such as: sol-gel phase transition, biocompatibility, sustained drug release, and in situ tumoral administration capable of promoting the best results in cancer therapy [33]. Our results demonstrated the VADE-loaded-hydrogel prepared with winter extract promoted a cytotoxic effect in tumor cells, suggesting the involvement of the polymeric micellar system in the cytotoxic effect of this herbal extract loaded in Pluronic^®^ F127. 

### 3.5. Evaluate of Thermosensitive Hydrogel Effect on Cell Morphology

SCC-25 and L929 morphological features were evaluated by Giemsa staining and transmission electron microscopy after a 24 h VADE-loaded-hydrogel incubation at different concentrations. The analysis of controls (SCC-25 without treatment—negative control; Figure 8A) and treated with vector control at an equivalent concentration to the higher concentration of VADE-loaded-hydrogel (Figure 8B), showed the absence of morphological alterations. However, SCC-25 cells treated at 250 µg/mL presented nuclear and cytoplasmic morphological changes (Figure 8C, arrows), such as: cellular rounding and swelling (arrow 1), vacuoles (arrow 2), chromatin condensation (arrow 3), chromatin fragmentation (arrow 4), and intensive cell lysis (arrow 5). These morphological changes were similar at the highest concentration of VADE-loaded-hydrogel (500 µg/mL), as well as a marked decrease in cell number (Figure 8D). 

The SCC-25 ultrastructural analysis of untreated cells showed apparently unaffected morphology with preserved plasmatic and nuclear membranes, and normal cytoplasmatic vacuoles (Figure 9A). However, after 24 h of incubation with VADE-loaded-hydrogel at 500 µg/mL important damage was observed, which could be strongly associated with cellular metabolism alteration detected by MTT assay. The most important changes included: disorganization of the cellular plasmatic membrane, increased vacuole size (Figure 9B, arrow 1), and alteration in the cristae mitochondrial shape (Figure 9B, arrow 2). Furthermore, intensive cell lysis with the generation of debris and amorphous cellular material was observed (Figure 9C). 

The morphological changes observed in SCC-25 cells by Giemsa and TEM suggest a necrosis process since cellular and mitochondrial swelling and lysis [66], increase in vacuoles, coagulation of chromatin [67], and its fragmentation were observed [68,69]. Furthermore, the alteration in the cristae mitochondrial shape can lead to the interruption of oxidative phosphorylation, and, therefore, cell death is triggered [70]. 

It has already been reported that viscotoxins (VT), a very well-known polypeptide present in *V. album*, constitute an important group of phytocompounds involved with the antitumoral potential of this semi-parasitic plant [23]. The high content of cysteine in the VT amino acid chain increases the positive charges of these cationic proteins [71]. The amino-acid-sequence and the tridimensional structure of these basic proteins, influence the net positive charge and the specific interaction of VT isoform with cellular involucre [72]. Holandino et al. [22] observed important differences in the viscotoxin contents when hydroalcoholic extracts of *V. album* subspecies were analyzed. They showed that *V. album* subsp. *abietis* growing on *Abies alba,* the same species evaluated in this study, had a higher content of VT A3, and a higher cytotoxic against Yoshida and MOLT-4 tumoral cell lines.

The isoform A3 (VT A3) is the most active VT since binds with high affinity to membranes negatively charged [73] as usually detected in tumor cells. Considering that the presence of glycoconjugates increases the negative charge of the tumor cellular surface [74], the electrostatic interaction of VTA3 with tumor cells could be responsible for the highest cytotoxic effect of VADE-loaded-hydrogel in SCC-25. Furthermore, Chen et al. [75] proposed that the negative charges on the surface of cancer cells are also associated with the increase of glycolytic metabolism and the increase of lactate anions secretion, which are well-known hallmarks of the tumorigenic process. On the other hand, the surface of normal cells remains charge-neutral because these cells present a normal glycolysis pathway. These findings could explain the tumor cell selectivity triggered by VADE-loaded-hydrogel in our in vitro experiments (Figure 7). 

The morphological and ultrastructural aspects registered in non-tumor L929 cells indicated the absence of toxic effects associated with VADE-loaded-hydrogel at different concentrations (Figure 10 and Figure 11). The Giemsa methodology indicated the absence of significant morphological alterations when control cells (Figure 10A,B) were compared to experimental groups (VADE-loaded-hydrogel at 500 µg/mL—Figure 10C; VADE-loaded-hydrogel at 1000 µg/mL—Figure 10D). 

A similar cellular profile was detected by transmission electron microscopy in which the absence of damage was detected in vital organelles of L929 cells (Figure 11A,B). The morphological study of tumor cells (SCC-25) and non-tumor cells (L929) confirmed the data obtained from cell viability, in which a higher cytotoxic potential for tumor cells (IC_50_ 333.4 µg/mL) was registered in comparison to normal ones (IC_50_ > 1000 µg/mL). Our results clearly show different cellular effects generated in vitro by VADE-loaded-hydrogel, highlighting the promising pharmaceutical application of Pluronic smart hydrogel as a drug delivery system to *V. album* dry extract. 

## 4. Conclusions

The use of thermosensitive hydrogel containing dry extract *V. album* (VADE-loaded-hydrogel) showed a promising anti-cancer potential detected by in vitro assays. The analysis conducted by DLS and transmission electron microscopy showed uniform micelles structures containing dry extract (VADE) homogenously dispersed. The DSC and TGA experiments indicated an increase in thermal stability induced by VADE. Flavonoid content quantified in the VADE-loaded-hydrogel showed similar amounts as previously detected in other studies, indicating reproducibility of the chemical results when different batches of dry extracts of *V. album* are analyzed. Bioactive phytocompounds, such as flavonoids and viscotoxins, are probably connected to the cytotoxicity effects, which were selective to tumoral cells. The use of Pluronic^®^ F127 hydrogel as a drug delivery system should also permit site-directed drug delivery, enhancing bioavailability. Therefore, the thermosensitive *V. album* hydrogel has the potential to become a therapeutic agent against solid tumors as an in situ gel because of its thermal stability at body temperatures, as registered in our experiments. Additional experiments are required to confirm the antitumoral potential of VADE-loaded hydrogel in in vivo complex organisms.

## 5. Patents

The developed formulation (VADE) was submitted in Brazil (Instituto Nacional de Propriedade Intelectual) requiring intellectual properties, under the number BR 10 2020 003600 9, on 20 February 2020.

## Figures and Tables

**Figure 1 pharmaceutics-14-02775-f001:**
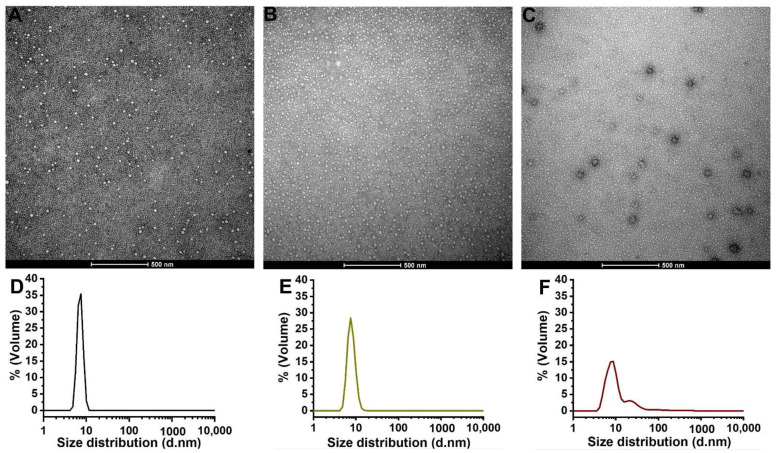
Representative images obtained by TEM (**A**–**C**) and distribution percentage by volume graphs of the micelles analyzed by DLS (**D**–**F**). The images and graphics are, respectively, from the following samples: (**A**,**D**) micelles formed in Pluronic^®^ F127 solution in water; (**B**,**E**) micelles formed from the hydrogel solution without VADE; and (**C**,**F**) micelles formed from VADE-loaded-hydrogel (1:10). The experiment was performed once and in triplicate. Scales bar = 500 nm.

**Figure 2 pharmaceutics-14-02775-f002:**
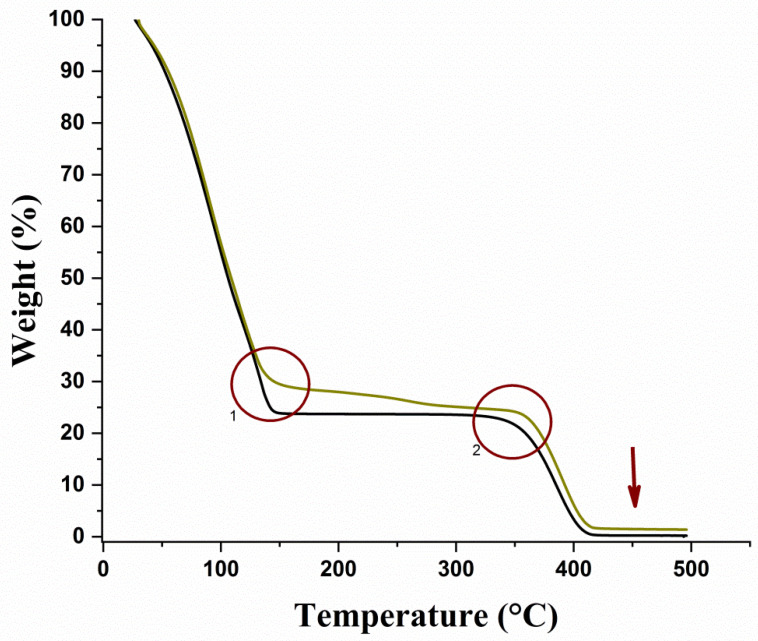
TGA curves of VADE-loaded-hydrogel (green curve) and without VADE (black curve) in the range from 30 to 500 °C. The first thermal degradation occurred at 30 °C to 143 °C and was attributed to water loss. The second stage started at 350 °C, and at 410 °C a total decomposition of Pluronic^®^ F127 was registered. Circle 1 highlights the difference in mass weights between VADE-loaded-hydrogel (green curve) and the sample without VADE (black curve). Circle 2 shows that the polymer decomposition occurred at different temperatures, starting at 346 °C in the absence of VADE, and at 365 °C in the presence of VADE. The arrow indicates the presence of mineral residuum in the VADE- loaded-hydrogel.

**Figure 3 pharmaceutics-14-02775-f003:**
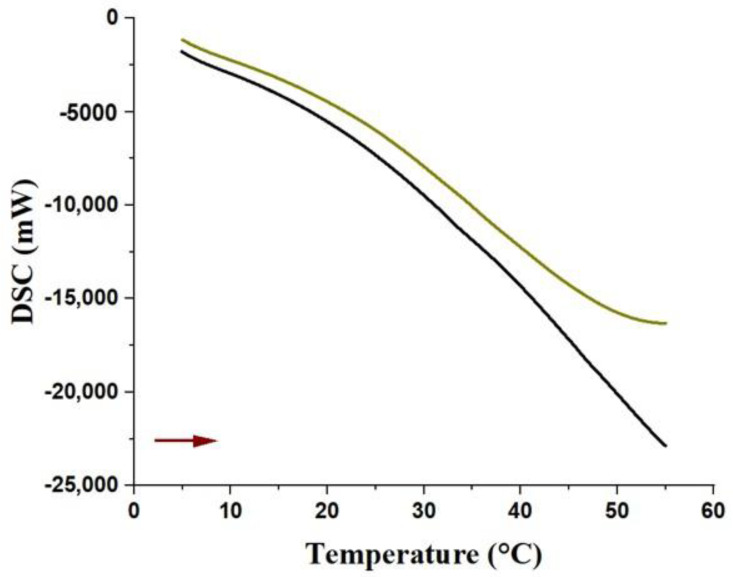
Thermogram of samples analyzed by DSC. Heat flux graph: heating versus temperature (0–60 °C; red arrow) of the hydrogel without VADE (black curve) and VADE-loaded-hydrogel (green curve).

**Figure 4 pharmaceutics-14-02775-f004:**
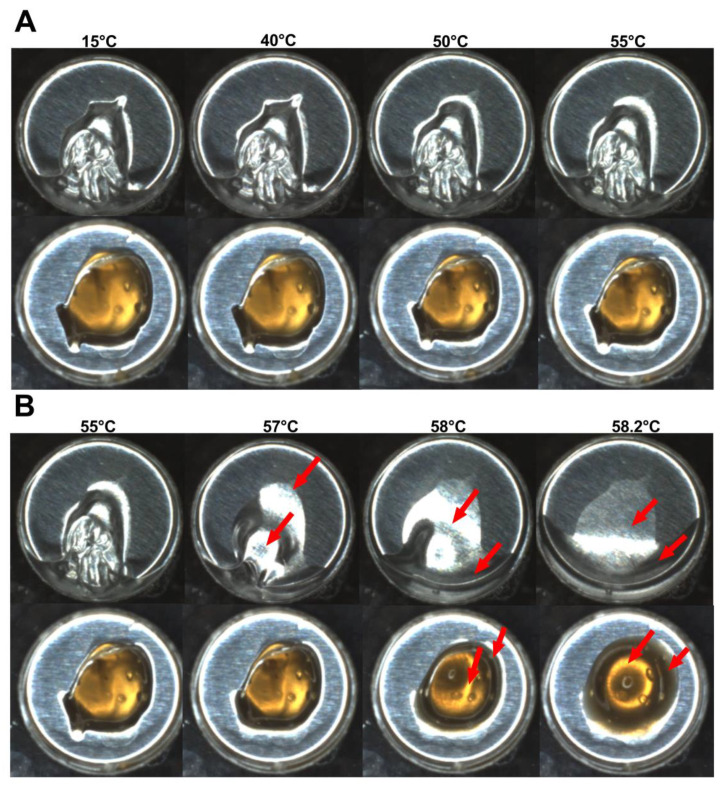
Sample images during the heated portion of the DSC analysis. Images of the following samples, respectively: (**A**) Hydrogel without (top panel) and VADE-loaded-hydrogel (bottom panel) are shown, respectively; from the left to the right at 15, 40, 50, and 55 °C. (**B**) Hydrogel without (top panel) and VADE-loaded-hydrogel (bottom panel) are shown, respectively; from the left to the right at 55, 57, 58, and 58.2 °C. The arrows show the main physical changes observed in the samples.

**Figure 5 pharmaceutics-14-02775-f005:**
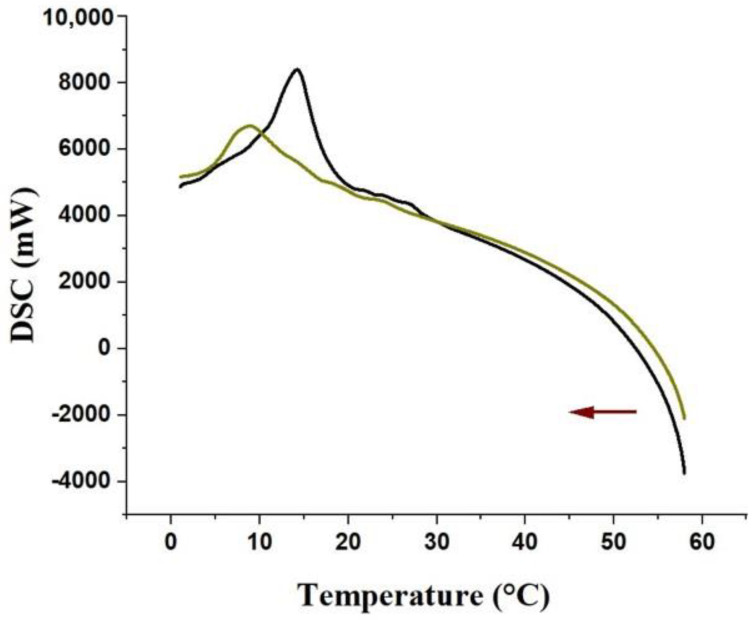
The heat flux graph during cooling versus temperature (60–0 °C; red arrow) of the VADE-loaded-hydrogel (green curve) and hydrogel without VADE (black curve).

**Figure 6 pharmaceutics-14-02775-f006:**
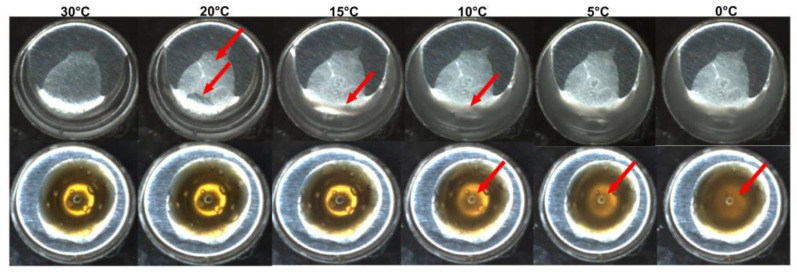
Respective images of hydrogel samples were recorded simultaneously by video during the DSC analysis. The range of temperature from the left to the right was: 30 °C, 20 °C, 15 °C, 10 °C, 5 °C, and 0 °C. The images are, respectively, taken from the following samples: hydrogel without VADE (**up-panel**) and VADE-loaded-hydrogel (**down-panel**). The arrows show the main physical changes observed in the samples.

**Figure 7 pharmaceutics-14-02775-f007:**
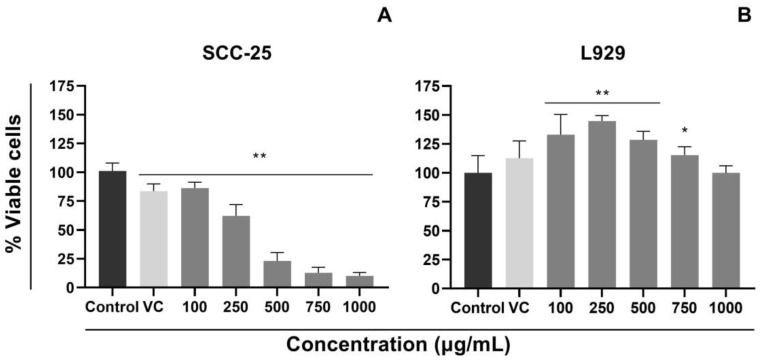
Cytotoxic effect of VADE-loaded-hydrogel on SCC-25 (**A**) and L929 (**B**) cell lines, after 24 h of treatment. Data were reported as mean ± SD of at least three independent experiments, done in quintuplicate. * *p* < 0.05; and ** *p* < 0.001, measured by analysis of variance (ANOVA), followed by Tukey post-test. VADE = *Viscum album* dry extract; Control = cells in culture medium (without treatment); VC = vector control (hydrogel without VADE).

**Figure 8 pharmaceutics-14-02775-f008:**
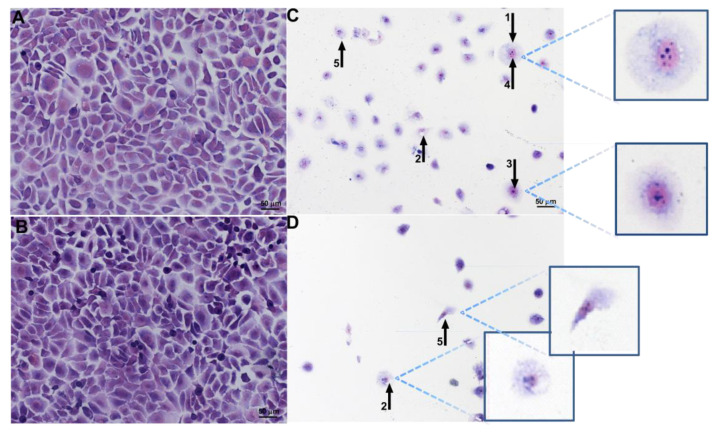
Morphological changes of SCC-25 cells assessed by Giemsa staining, after 24 h of treatment. (**A**) Negative control; (**B**) Vector control (hydrogel without VADE) at higher concentration; (**C**) VADE-loaded-hydrogel at the concentration of 250 µg/mL and (**D**) 500 µg/mL. The arrows and insets in (**C**,**D**) highlight the most important cellular morphological alterations, such as: 1—rounding and swelling; 2—vacuoles; 3—chromatin condensation; 4—chromatin fragmentation; 5- lysis. The intensive cellular lysis was detected at a higher concentration of VADE-loaded-hydrogel (**D**). Scale bars = 50 µm.

**Figure 9 pharmaceutics-14-02775-f009:**
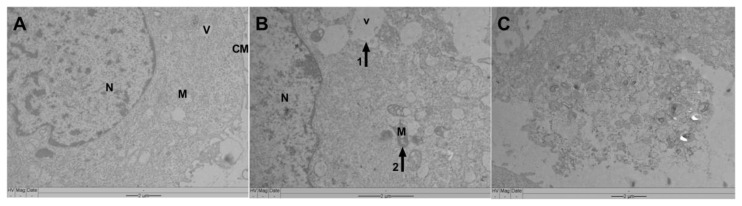
Morphological changes of SCC-25 cells assessed by transmission electron microscopy, after 24 h of treatment. (**A**) SSC-25 without treatment (control); (**B**) SCC-25 treated with 500 µg/mL of VALE-loaded-hydrogel (arrow 1: cytoplasmatic vacuole; arrow 2: mitochondria damage); (**C**) SCC-25 treated with 500 µg/mL of VALE-loaded-hydrogel: intensive cell lysis. M: Mitochondria; V: vacuole; CM: cytoplasmatic membrane; N: nucleus. Scale bars: 2 µm.

**Figure 10 pharmaceutics-14-02775-f010:**
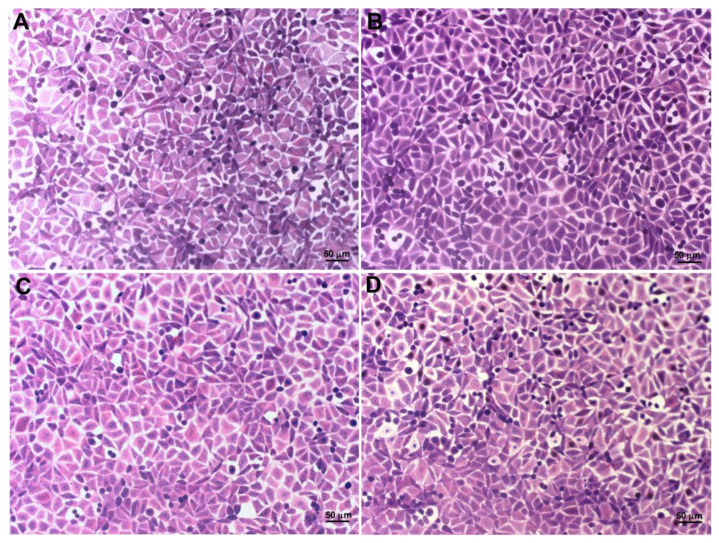
Morphological aspects of L929 cells were assessed by Giemsa staining, after 24 h of treatment. (**A**) Negative control; (**B**) Vector control (hydrogel without extract) at higher concentration; (**C**) VADE-loaded-hydrogel at 500 µg/mL; (**D**) VADE-loaded-hydrogel at 1000 µg/mL. Scale bars = 50 µm.

**Figure 11 pharmaceutics-14-02775-f011:**
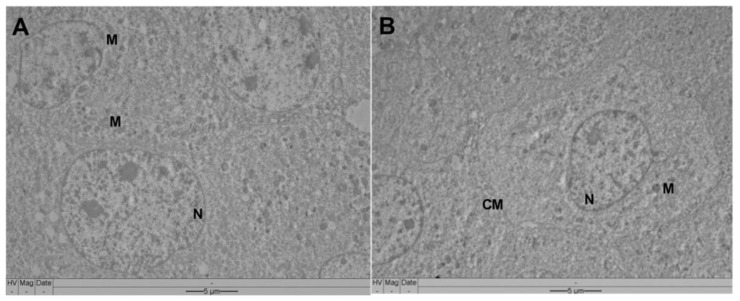
Morphological changes of L929 cells were assessed by transmission electron microscopy, after 24 h of treatment. (**A**) Negative control; (**B**) VADE-loaded-hydrogel at 500 µg/mL of extract. M: Mitochondria; N: nucleus; CM: cytoplasmatic membrane. Scale bars: 5 µm.

## Data Availability

Not applicable.

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
