# Peer review of "Pluronic® F127 Thermoresponsive Viscum album Hydrogel: Physicochemical Features and Cellular In Vitro Evaluation"

_pharmaceutics, 2022, doi:10.3390/pharmaceutics14122775_

Round 1
Reviewer 1 Report
The presented paper is devoted to interesting topic - application of extracts from plant as anticancer agent. The anticancer effect and selective toxicity of formulation towards tumor cells as compared with normal fibroblasts was shown. However, there are few major questions, which should be met by authors before publication of this paper.
1. In order to provide relevance to pharmacological area the data on VADE chemical composition should be provided. In current form the results are completely not reproducible. Please, add information in Introduction on the active molecules, which are components of the VADE, and mechanism of their anti-cancer action. This should provide clear information for the readership. Also, please provide the chemical composition of VADE, which you have applied in your research. The IC50 concentrations should be reffered to the active molecule. However, authors do not provide the information on CONCENTRATION OF WHICH SUBSTANCE they have used for cytotoxicity evaluations. This information is strongly needed for understanding of results!!!
2. How reproducubility of biological effect (anticancer activity) could be retained? In my opinion the active substance or their combination should be determined and characterized. The quntities of such could differ from batch to batch. However, when dealing with known substance, but NOT JUST DRY EXTRACT, it is possible to correct the composition. Pharmaceutical application should start from active molecule and possible mechanisms of action. Otherwise it is not scientifically sound!
3. The information on pages 6-12 is mostly about physico-chemical properties, but no relevance to pharmaceutical properties and inportance of these data for pharmaceutical application was provided. At the same time it ia not clarified by the authors - what is the real formulation. Is it miccelar solution, which should turn to gel in situ, or it is a gel itself. Please, provide the physico-chemical propertiea as related to pharmacological ones.
4. The composition of the gel was not provided. How much of the formulation is polymer and how much is of the drug (dry extract)? Drug loading and dissolution data should be provided. It will be also useful to see some scheme, describing the formulation and its properties. Micells and gel - how they organazed in the structure? On the photos and videos we can see just macro gel. However there are micells on the TEM photos.
Minor comments:
1. Line numbers were not provided.
2. Abstract. Not Viscum album, but extract from it possess anticancer activity. Please correct.
3. Abstract. There is no need to use Pluronic F-127 to solubilixe hydrophilic compounds. These are soluble on their own.
4. Figs 4 and 6. Please show the effect you want to show with arrows. The effect is not clear.
5. Page 10, "...second involves the conformation of the polymer chain based on possible rotations..." Please explain how this could affect cloud point.
6. Page 17. What is the reason and possible mechanism for such great selectivity. Please add information.
Please see the attache file withe comments of the reviewer.

Reviewer 2 Report
The Authors investigated the physicochemical features of Viscum album dry extract loaded in a polymeric delivery system of Pluronic® F127 (VADE-loaded-hydrogel) through various techniques, among them transmission electron microscopy. They also evaluated the anti-tumor potential of the hydrogel through in vitro assays on cancer cells (oral tongue human squamous carcinoma - SCC-25 cell line) and normal mouse adipose fibroblasts (L929 cell line). The morphological cellular aspects triggered by VADE-loaded-hydrogel were assessed by optical and electron microscopy. They found that the presence of the extract in the hydrogel increased its thermal stability, while having a similar decomposition curve with the simple hydrogel. The mistletoe extract showed tumor selectivity as indicated by the IC50 dose which was much lower for cancer cells, compared to normal cells. Additionally, effects such as plasmatic membrane disorganization, increase of cytoplasmic vacuoles` size, alteration in the cristae mitochondrial shape, and generation of amorphous cellular material, were only detected in the squamous carcinoma cells, reinforcing tumor specificity of the VADE-loaded-hydrogel. The manuscript is generally well written and the workflow is adequately described, therefore the Reviewer believes the manuscript will have an impact in the field considering the need for reliable delivery systems for herbal non-aqueous drugs with anti-cancer potential.
However, the Authors should amend the following aspects before the manuscript becomes suitable for publication:
- - The Authors mention they used mouse normal fibroblasts (L929 cell line isolated from subcutaneous areolar and adipose tissue) and oral human squamous cell carcinoma (SCC-25 cell line isolated from the tongue) to perform IC50 evaluation for VADE and determine the selectivity index (ratio between the IC50 of non-tumor cells and IC50 of cancer cells), however such comparison should be made between normal and cancer cells from the same species and same type of tissue in order to obtain relevant results.
- The Authors used VADE-loaded-hydrogel dissolved in full medium at the following concentrations of VADE: 100, 250, 500, 750 and 1,000 µg/mL for IC50 evaluation (as stated in page 4, section 2.9 and the results shown in Figure 7). However, they only mention the IC50 values for tumor cells and normal cells to be 333.40 µg/ mL and 1,433 µg/ mL respectively (abstract and page 13); how did they reach these values? How did they choose these concentrations?
- -The Reviewer encourages the Authors to showcase the significant morphological changes highlighted in Figure 8C using magnification, as they are not easily observed by the reader.
- -The images shown in Figure 11 should have the same magnification and scale bar in order to be comparable.
- - The Authors are advised to spellcheck their manuscript with a native English speaker.
Round 2
Reviewer 1 Report
Authors have significantly improved the Manuscript and introduced several important references, and discussion. In my opinin paper could be accepted for publication.